# Biogenic Synthesis of Silver Nanoparticles Using *Catharanthus roseus* and Its Cytotoxicity Effect on Vero Cell Lines

**DOI:** 10.3390/molecules27196191

**Published:** 2022-09-21

**Authors:** Khansa Jamil, Sahir Hameed Khattak, Anum Farrukh, Sania Begum, Muhammad Naeem Riaz, Aish Muhammad, Tahira Kamal, Touqeer Taj, Imran Khan, Sundus Riaz, Huma Batool, Kaleemullah Mandokhail, Sabahat Majeed, Sajid Ali Khan Bangash, Alia Mushtaq, Shahab Bashir, Imdad Kaleem, Fahed Pervaiz, Aamir Rasool, Muhammad Ammar Amanat, Ghulam Muhammad Ali

**Affiliations:** 1National Institute for Genomics and Advanced Biotechnology (N.I.G.A.B.), National Agriculture Research Centre (NARC), Islamabad 44000, Pakistan; 2Department of General Medicine, Fauji Foundation Hospital, Rawalpindi 45000, Pakistan; 3Ecotoxicology Department, National Agriculture Research Centre (NARC), Islamabad 44000, Pakistan; 4Food Sciences Research Institute, National Agriculture Research Centre (NARC), Islamabad 44000, Pakistan; 5Department of Botany, Sardar Bahadur Khan’s Women’s University, Quetta 87300, Pakistan; 6Department of Microbiology, Balochistan University, Quetta 87300, Pakistan; 7Department of Biosciences, COMSATS University, Islamabad 45550, Pakistan; 8Institute of Biotechnology and Genetic Engineering (IBGE), Khyber Pakhtunkhwa Agriculture University, Peshawar 25130, Pakistan; 9Crop Disease Research Institute CDRI, National Agriculture Research Centre, Islamabad 44000, Pakistan; 10Institute of Biochemistry, Balochistan University, Quetta 08770, Pakistan

**Keywords:** nanoparticles (NPs), silver nanoparticles (AgNPs), *C. roseus*, Vero cell line, animal model, antidiabetic potential

## Abstract

**Background**: Type 2 diabetes mellitus (DM2) is a chronic and sometimes fatal condition which affects people all over the world. Nanotherapeutics have shown tremendous potential to combat chronic diseases—including DM2—as they enhance the overall impact of drugs on biological systems. Greenly synthesized silver nanoparticles (AgNPs) from *Catharanthus roseus* methanolic extract (*C.* AgNPs) were examined primarily for their cytotoxic and antidiabetic effects. **Methods**: Characterization of *C.* AgNPs was performed by UV–vis spectroscopy, Fourier transform infrared spectroscopy (FTIR), X-ray diffraction (XRD), and atomic force microscopy (AFM). The *C.* AgNPs were trialed on Vero cell line and afterwards on an animal model (rats). **Results**: The *C.* AgNPs showed standard structural and functional characterization as revealed by FTIR and XRD analyses. The zetapotential analysis indicated stability while EDX analysis confirmed the formation of composite capping with Ag metal. The cytotoxic effect (IC50) of *C.* AgNPs on Vero cell lines was found to be 568 g/mL. The animal model analyses further revealed a significant difference in water intake, food intake, body weight, urine volume, and urine sugar of tested rats after treatment with aqueous extract of *C.* AgNPs. Moreover, five groups of rats including control and diabetic groups (NC1, PC2, DG1, DG2, and DG3) were investigated for their blood glucose and glycemic control analysis. Conclusions: The *C.* AgNPs exhibited positive potential on the Vero cell line as well as on experimental rats. The lipid profile in all the diabetic groups (DG1-3) were significantly increased compared with both of the control groups (*p* < 0.05). The present study revealed the significance of *C.* AgNPs in nanotherapeutics.

## 1. Introduction

The prevalence of type 2 diabetes mellitus (DM2) and the corresponding demand for its management are proving to be a global economic burden [1]. Various strategies, including dietary management, exercise-based therapies, novel drug development, homeopathic and herbal medicines, pharmacotherapy, nanotechnology, traditional medicines, and combined anti-diabetic or ‘anti-hyperglycemic’ therapies are prioritized as various methods for treatment and prevention of DM2 globally [1]. In recent years, various target-oriented scientific approaches have been devised to overcome the loss inflicted by lethal diseases of plants and animals [2]. Among them, nanotherapeutics have proven to have tremendous potential to combat chronic/fatal diseases as they enhance the overall impact of drugs on biological systems by increasing absorption of drug, improving drug solubility, extending drug half-life, raising a drug’s therapeutic index, and reducing the immunogenicity of a drug [2]. Nanotherapeutics encompasses the development, distribution, monitoring, and control of drugs in terms of repair, defense, and improvement of human biological systems at the molecular level. This emerging multidisciplinary field has a wide range of applications [3,4,5]. The utilization of biological systems such as plants, animals, bacteria, and other organisms in nanomaterial production is becoming common practice [6]. Various types of nanoparticles and their characterization and applications can be employed in various sectors, particularly in the health and chemical industries [7]. The applications of nanoparticles in cosmetics, pharmaceuticals, and agrochemicals are enormous [8], therefore, various approaches—including physical and chemical methods—have been used to synthesize NPs [9]. Green synthesis has proven to be advantageous due to its reliability, eco-friendliness, cost-effectiveness, and sustainability [9,10,11]. The plant entities used as in production and assembly of AgNPs have gained attention due to their quick, non-hazardous, eco-friendly, and economical protocol, as they provide a simple, single-step technique for the biosynthesis of nanoparticles [12]. Finding novel hypoglycemic medications with fewer side effects along with better effectiveness is a worthwhile endeavor, and this has been the focus of previous study [13]. Biologically inspired green synthesis of nanoparticles has become a promising alternative against the chemically fabricated process considering reduced toxicity [14]. The *C. roseus* belongs to the Apocynaceae family and originated in Madagascar. It is highly medicinal and its terpenoid indole alkaloids are greatly effective in fighting cancer and diabetes [15,16]. The medicinal constituents of *C. roseus* have made it a highly valuable plant and its exploitation in nanotherapeutics has been investigated in the present study. Many studies were unable to present the dose-dependent toxicity data on important cell lines and different types of diabetes [6,7,8,9,10,11,12]. Although anti-diabetic and hypoglycemic medicines are used to treat diabetes [17,18], their long term use can cause various complications [19]. Therefore, non-toxic, economical, and easily available plants possessing antidiabetic properties—such as *C. roseus*—have been selected and utilized in this study. There are many studies on the biosynthesis of silver nanoparticles (AgNPs), but only a few have described the green synthesis of AgNPs from *C. roseus*; however, no report in literature is available on the cytotoxic and antidiabetic effects of *C.* AgNPs [20,21,22,23,24,25,26,27,28]. Here, we report the greenly synthesized silver nanoparticles (AgNPs) from methanolic extract of *C. roseus* and designated as (*C.* AgNPs). The characterization and nanotherapeutic effects of *C.* AgNPs were investigated for cytotoxic and antidiabetic activities. The cell viability or cytotoxic effects of *C.* AgNPs were determined by their employment on Vero cell lines. The antidiabetic properties of *C.* AgNPs were examined on mammalian model (rats). All-important antidiabetic pararmeters—including blood glucose, anti-hyperglycemic analyses, water intake, food intake, body weight, urine volume, urine sugar, and lipid profile—were investigated thoroughly. The present study encompasses the unique data in terms of cytotoxic and antidiabetic effects of *C.* AgNPs.

## 2. Materials and Methods

### 2.1. Collection of Plant Sample

*C. roseus* leaves were collected from the N.I.G.A.B., NARC greenhouse and identified by the Plant Taxonomy Laboratory of Quaid-e-Azam University in Islamabad. Clorox (10 percent) was used to disinfect the *C. roseus* samples for 10 min to eliminate dirt and dust. Before processing, they were cleaned in distilled water and dried in the shade for 3 to 4 weeks at room temperature, powdered, and then sieved through a 20 mm mesh with an electric mixer grinder (Panasonic, MX-AC300) into homogenous granules. Using a rotary evaporator (BUCHI B-480) set to 45 °C, the samples were cooled to ambient temperature in zip-top reusable plastic bags before being re-suspended in new methanol [24]. For 10 min, the mixture was spun at 14,000 rpm in a Whatman No.1 filter before it was centrifuged in the same manner. Vacuum evaporation was used to remove the remaining methanol before microfiltration with 0.20 mm cellulose membrane filters (Millex, Bedford, MA 01730, USA). For future usage, the filtrate was collected in falcon tubes and kept at a temperature of 4 °C in the freezer.

### 2.2. Preparation of Plant Extract

Methanol was used to dissolve 30 g of dry powder, which was kept at room temperature in a rotating shaker for three days (V-TECH-HZ 300). Re-suspended in new methanol, the residue was filtered using Whatman No.1 filter paper for three days, after which the mixture was purified. A rotary evaporator (BUCHI B-480) was used to evaporate methanol under a vacuum at 45 °C and then re-suspended in new methanol. After being centrifuged for 10 min at 14,000 rpm, the mixture was filtered using Whatman No.1 filter paper. Vacuum evaporation was used to remove the remaining methanol before microfiltration with 0.20 mm cellulose membrane filters (Millex, Bedford, MA 01730, USA). For future usage, the filtrate was collected in falcon tubes and kept at a temperature of 4 °C in the lab in a rotary shaker [24].

### 2.3. Green Synthesis of Silver Nanoparticles

In a 100 mL Erlenmeyer flask, 10 mL of *C. roseus* leaves extract were combined with 90 mL of a 1 mM silver nitrate aqueous solution and stirred at 60 °C for one hour. The same conditions were used to maintain a negative control that contained only silver nitrate aqueous solution. The UV–visible spectrum was recorded and the color shift was observed regularly to track the process’ progress. *Cathranthus* silver nanoparticles were detected in the positive control, which changed color from pale yellow to greenish-brown. No color change was detected in the negative control. Excess Ag ions and contaminants were removed from the solution by centrifuging it at 14,000 RPM for 20 min after the reaction was complete. The resulting pellet was then cleaned with deionized water. The centrifugation and washing process was repeated three times to efficiently separate the nanoparticles. A 55 °C oven was used to dry the silver nanoparticles derived from the *C. roseus* plant.

### 2.4. Characterization of Silver Nanoparticles (AgNPs)

This study used Fourier Transform Infrared (FT-IR), Atomic Force Microscopy (AFM), TESCAN microscopy, SEM EDAX, Zetapotantial, X-ray Diffraction (XRD), Zetasizer/Dynamic light scattering, and UV spectroscopy to characterize silver nanoparticles in the early stages of their development.

### 2.5. Fourier Transform Infrared (FT-IR) Spectroscopy Analysis

It was necessary to disperse silver nanoparticles from *Cathranthus* on either a Si plate or a Ge A.T.R. crystal before drying to produce FTIR spectra of the investigated samples. We obtained the spectra, averaging 64 interferograms with a resolution of 400 cm^−1^. An H.P. UV 8543 diode-array spectrometer was used to gather the extinction UV–vis spectrum of Ag NPs in cyclohexane with a resolution of 1 nm [29].

### 2.6. Atomic Force Microscopy (AFM)

We used an Agilent 5500 Ar AFM (Agilent, Santa Clara, CA, USA) with a tap mode to accomplish the AFM imaging (ACAFM). A silicon nitride cantilever was used for contact mode imaging (NP-20, nominal frequency 56 kHz, nominal spring constant 0.32 N/m: M.S.N.L., nominal frequency 4–30 kHz, nominal spring constant 0.010–0.070 N/m). The force was kept at the lowest possible value to limit the forces of interaction between the tip and the surface. A resonance frequency range of 289–335 kHz and a spring constant of 20–80 N/m were used for the tapping mode. The scanning frequency was set at 1.01 Hz per line, and the AFM pictures were taken in a weightless chamber with a scanning range of 2.0 to 2.5 [29].

### 2.7. TESCAN Microscopy and SEM EDX

Scanning electron microscope MAIA3 TESCAN-ultra high-resolution SEM with Schottky field emission cathode was used for electron micrographs. Images were taken using a combination of InBeam SE + low energy BSE detector at 20 kV. Furthermore, the product morphology was also observed by scanning electron microscope and EDS analysis were performed using the APOLLO X analyzer (EDX) [29].

### 2.8. Zeta-Potential and Zetasizer/Dynamic Light Scattering (DLS)

The *C.* silver nanoparticles were sonicated for 20 min and then adjusted to 75 μg/mL concentration in phosphate buffer saline (PBS pH = 7.4). The zeta potential of the diluted *C.* silver nanoparticles solution was then estimated using a Zetasizer (D.L.S., 4.9) [29]. The Zetasizer/dynamic light scattering (DLS) measurements were carried out using Malvern instruments [29].

### 2.9. Cytotoxicity Assay

The synthesized *C.* AgNPs were tested against Vero cell lines in vitro for cytotoxicity. Dulbecco’s modified eagle medium (DMEM) supplemented with 10% F.B.S. and 1% Penstrep was used to cultivate frozen cells (at a concentration of 4 × 105 cells per well) at 37 °C in humidified air containing 5% CO_2_ at 37 °C. The medium was removed to eliminate any dead cells, and the cells were rinsed with PBS. Serial dilutions of *C.* AgNPs, with a range of 142–1000 g/mL, were incubated with Vero cell lines cultivated in microplates. Plates were also made for the positive and negative controls. Cell viability and cytotoxicity index IC_50_ values were calculated after 48 h of incubation at 37 °C in a 5 percent CO_2_ incubator [30,31,32,33,34,35].

### 2.10. Experimental Design

Fifty SPF-grade male Wistar rats (140 ± 15 g, 6 weeks old), with a body weight of 160 ± 10 g after 10 days of accommodation, were used for the experiment (Ethical approval no. 528/Reg./MICB/2022). The DM2 rat model was established according to the standard protocol (The State Food and Drug Administration, 2012) with minor modifications. After an accommodation period of 10 days, all the rats (except 10 rats set as a normal control group) were kept on fast for 24 h (free drinking water) and injected intraperitoneally with streptozotocin (STZ) at a dose of 38 mg/kg body weight (b.w.). The STZ is dissolved in 100 mM citrate buffer (cold) at pH 4.5 before use. Five days after STZ induction, the blood samples were collected from the tails of the rats (5–6 h fast). Rats were randomly divided into five groups as follows: Group I normal control (NC1): normal rats receiving sterile distilled water (*n* = 10); Group II positive control (PC2): normal rats treated with 510 mg/kg/day aqueous *C. roseus* extract (*n* = 10); Group III (DG1): diabetic rats receiving sterile distilled water (*n* = 10); Group IV(DG2): diabetic rats treated with 500 mg/kg/day of silver nanoparticles with aqueous *C. roseus* extract (*n* = 10) and group V (DG3): diabetic rats treated with commercially available medicine (Getryl, 2 mg once a day). The aqueous extract of silver nanoparticles with aqueous *C. roseus* was dissolved in sterile distilled water and administered intragastrically for 8 weeks.

During the experimental period, food and water intake of each cage were weighed daily, weight was measured weekly, and fasting blood glucose (FBG) was measured from the tail vein of the 5–6 h fasted rats every week. The oral glucose tolerance test (OGTT) was performed according to the method of The State Food and Drug Administration (2012). For OGTT, the rat blood samples were collected and determined at 0, 30, and 120 min of orally administered 2 g/kg glucose solution. The value of the area under the curve (AUC) was computed as follows: AUC = 0.5 × (G 0 h + G 0.5 h) × 0.5 + 0.5 × (G 2 h + G 0.5 h) × 1.5. At the end of the experiment, rats were given anesthesia with 10% chloral hydrate (0.3 mL/100 g), and then the blood samples were withdrawn from the abdominal aorta of rats. Afterwards, collected serum was centrifuged at 2500 rpm at 4 °C for 12 min, and stored at −20 °C for total cholesterol (TC), triglycerides (TG), low density lipoprotein cholesterol (LDL-C), high density lipoprotein cholesterol (HDL-C), and serum insulin analysis.

## 3. Results

### 3.1. Synthesis of C. roseus Silver Nanoparticles (AgNPs)

Silver nanoparticle biogenic synthesis was based on *C. roseus*. A change in color from pale greenish yellow to dark brown was the first sign that silver nanoparticles had been synthesized and confirmed by ocular examination. Biogenically produced silver nanoparticles have been extensively characterized using UV–visible spectra. As a lowering agent, phytochemicals were utilized (Figure 1).

### 3.2. Characterization of C. AgNPs

#### 3.2.1. Fourier Transform Infrared (FT–IR)

Fourier transform infrared (FT–IR) measurements were carried out to identify the functional groups of pure *C. roseus* extract and extract of *C. roseus* silver nanoparticles shown in Figure 2. FT-IR spectrum of the experimental sample revealed stretching in the wavelength range of 4000–450 cm^−1^. The formation and stability of the reduced silver nanoparticles in the colloidal solution was monitored by FT–IR. The FT–IR spectra showed different peaks for the presence of functional groups in pure *C. roseus* extract at 890, 1000, 1360, 2850, 3000, and 3300 cm^−1^. After reaction with AgNo_3_, the peak shifted to the higher wave number side, such as when peak 1360 shifted towards 1800 cm^−1^, peak 2800 and 2900 in pure *C. roseus* stretched and shifted to the lower side observed at 2600 cm^−1^ which corresponds to the presence of C-H stretching bond. The intense peak at 3500 cm^−1^ indicated the presence of amide bonding. These functional groups are enlisted in (Table 1). The synthesis of *C. roseus* silver nanoparticles occurred by bonding Ag^+^ with an amide group present at the 3300–3500 cm^−1^ peak of pure extract. The immediate reduction and capping of silver ions into silver nanoparticles in the present analysis is due to presence of amide group in *C. roseus* extract that act as powerful reducing agents which may be suggestive of the formation of AgNps by reduction of silver nitrate.

#### 3.2.2. X-ray Diffraction (XRD)

Furthermore, evidence for the biosynthesis of silver nanoparticles was confirmed by the X-ray diffraction (XRD) analysis (Figure 3). The XRD analysis explained *C.*-based silver nanoparticles. In Figure 3, XRD patterns for silver nanoparticles revealed a distinct peak at 2θ ranging from 20° to 80°. The XRD confirmed that silver nanoparticles are crystalline with the face-centered cubic lattice. The XRD graph was plotted by intensity vs. 2θ. In the spectrum (Figure 3), the sharp peaks show the crystalline form in which atoms were arranged in a periodic pattern of three dimensions (3-D). The variation of peaks in the XRD spectrum was due to doping or foreign atoms capping so it confirmed the synthesis of the composite. The sharp peaks in the spectrum were crystalline and larger particles, while the small broader peaks were amorphous structures and smaller particles. Moreover, the high intensity of reflection (111) relative to the other as shown in the figure below indicates the silver nanocrystals as these peaks are mainly oriented along the (111) plane.

#### 3.2.3. Atomic Force Microscopy (AFM)

The results of atomic force microscopy (AFM) of *C.* AgNPs are shown in (Figure 4 and Table 2). The nanoparticles were examined using AFM in both surface and three-dimensional views, and the average size of *C.*-based silver nanoparticles was found to be 45 nm.

#### 3.2.4. Zetasizer and Zetapotential

The results of zetapotential and zetasizer (dynamic light scattering, DLS) are shown in Figure 5a. The zetapotential value was determined as −19.2 mV, a measure of the stability of the nanoparticles and mention the DLS value as 35 ± 10.5 nm. The negative value indicated the stability of the silver nanoparticles, and it evaded the accumulation of nanoparticles [36]. The magnitude of the zeta potential gives an indication of the potential stability of the colloidal system. If all of the nanoparticles in the suspension have a large negative zeta potential, then they will tend to repel each other and there will be no tendency for the particles to come together

The DLS size distribution image of biosynthesized silver-nano particles is shown in Figure 5b. It was observed that the size distribution of AgNPs ranges from 10 to 150 nm. The calculated average particle size is 35 nm ± 10.5. The zeta potential of the biosynthesized AgNPs was found as a sharp peak at −19 mV in the figure below. It was shown that the surface of the nanoparticles is negatively charged and dispersed in the medium. The negative charge confirms repulsion among particles and stability.

The stability of the synthesized *C. roseus* AgNPs recorded over 6 months showed no aggregation or any significant change in the surface plasmon resonance (SPR). Thus, the synthesized AgNPs were stable and could, therefore, be useful for various biomedical applications (Figure 5). In addition to this, we have also analyzed the DLS and zeta potential, in order to know the stability and charge of the synthesized AgNPs. Zeta potential is an important parameter that represents the surface charge of NPs, predicting interactions between NPs, and it is useful to foretell the long-term stability of NPs in suspension. The literature survey suggests that the NPs with higher negative zeta potential will repel each other and show high degrees of stability.

#### 3.2.5. SEM EDX (Scanning Electron Microscopy with Energy Dispersive X-ray Analysis)

To confirm the formation of composite capping with Ag metal EDX analysis was performed. During the EDX measurement, different areas were focused and corresponding peaks are shown in Figure 6a,b. Ag metal can be seen in the synthesized composite nanostructure in the EDX spectrum. The quantity of AlK and AgL was 73.34, and 26.6 respectively. The letters K and L are atom shells that refer to *n* value that electrons in that shell have. Atomic % values for AlK and AgL were 91.66 and 8.34 respectively and are listed in given Table 3. The EDX spectrum also revealed the AgNPs synthesis. The composition of AgNPs elements analyzed Al and Ag by the spectrum. Metallic silver nanocrystals with optical absorption peaks were approximately at 1.3 keV for Al and 2.8 keV for Ag respectively, due to surface plasmon resonance.

#### 3.2.6. TESCAN Electron Microscope

TESCAN image of the silver nanoparticle is shown in Figure 7. The surfaced morphology of silver nanoparticles (with SEMHV: 20.0 kV, view field 1.49 μm, SEM MAG 136 kx WD 8.79 mm) shows the even shape and spherical nature of the developed particle. This proves the successfulness of the developed particle that was further used for the model study and to check the effectiveness of the rat trial.

### 3.3. Cell Cytotoxicity Assay

The cytotoxicity of *C.* AgNPs was assessed using a viability cell assay on normal Vero cells to assess their safety for use in the medical field. The current findings revealed that low concentrations of *C.* AgNPs solution ranging from 142 to 568 μg/mL did not reduce cell viability. However, cell viability was reduced when higher concentrations of *C.* AgNPs solution from above 568 μg/mL were utilized. IC_50_ (the half-maximal inhibitory concentration) value of the *C.* AgNPs was 568 μg/mL. The analysis was also conducted to find out the best NP concentration at which there is less cell toxicity and more cell proliferation (Figure 8). Furthermore, one-way ANOVA (Table 4) revealed highly significant differences (*p* < 0.01) were observed between the concentration and the best concentration was 426 μg/mL with cell toxicity and cell proliferation at 37% and 70%, respectively. Hence it is clear from the result that the best nanoparticle that can be used to cure is concentration A3, followed by A4 with cell toxicity of 53% and cell proliferation of 60% (Figure 9).

The graphical representation of the data (Figure 9) indicates that the cytotoxicity and cell proliferation are directly proportional to each other until concentration A3, but inversely optional afterwards. Furthermore, the concentration below A1 and above A8 is not included as they had non-significant variations with poor cell proliferation. The point where both the graph lines (% cytotoxicity and cell proliferation) meet or the point where there is 50% inhibition of cell growth is considered as IC_50_ (in the present study value 568 near 50% hence considered as IC_50_).

### 3.4. Effect of the Aqueous C. roseus Extract on Water Intake, Food Intake, Rat Weight, Urine Volume, and Urine Sugar

The growth indicators as depicted in Figure 10a–d, the average water intake of NC1, PC2, DG1, DG2, and DG3 groups were significant (*p* < 0.05) with values of 26.41, 22.45, 156.96, 139.56, and 147.48 g respectively. Compared with the DG1 group, the water intake of the DG2 and DG3 groups decreased significantly in all the weeks apart from weeks 5 and 6 (for DG3). Similarly, the average food intake of NC1, PC2, DG1, DG2, and DG3 groups were highly significant (*p* < 0.01) with values of 22.31, 20.50, 42.87, 39.92, and 40.68 g respectively. Food intakes of diabetic rats were also significantly higher (DG1, DG2, and DG3 vs. NC1 and PC2) than in normal rats. Compared with the DG1 group, the food intake in the DG2 and DG3 groups decreased significantly (*p* < 0.01).

The weight of rats also varied significantly from weeks 2–8 of the experiment. The comparison of the groups revealed that the diabetic group DG1 gained more weight than other groups (Figure 10c). Similarly, DG2 had also the least increase in weight when compared with DG3, showing the significance of nanomedicine developed from plant extract. As shown in the figure, the urine volume of NC1, PC2, DG1, DG2, and DG3 ranged between 5.66 and 16.91 mL/d. The urine volume of diabetic rats (DG1, DG2, and DG3 group) was significantly increased to 16.92–17.28 mL/d, which was significantly higher (three times) than the normal group. Similarly, the urine sugar (Table 5) also varied significantly among the diabetic groups. Both the control groups showed no signs of sugar while the levels differed significantly between the DG1, DG2, and DG3 groups. The diabetic group without medication—i.e., DG1—showed a moderate to high level of sugar, while DG3 proved to be better than DG2 proving the effectiveness of commercially available medicine over nano-treatment. As far as nano-extract is concerned, weeks 2, 3, 7, and 8 showed better control of blood sugar, showing the effectiveness of the extract. The nano-extract given to the rats at 1, 4, 5, and 6 weeks showed a mild level of blood sugar control, proving the success of the trial. Though acute doses were given, on some occasions the nano-extract was not possible to control blood sugar properly but it still played its role as it performed better than the DG1 group. Furthermore, the presence of electrons (negative surface charge which may be attributed to deprotonated OH groups) at the silver nanoparticle surface which became more significant with reducing the particle size made them negatively charged. Nanoparticles with negative charge are less toxic in in vivo animal models as compared with the positive charged particles.

### 3.5. Glucose Tolerance in Hyperglycemia Model Animals

The results of glucose tolerance after four weeks are shown in Table 6. Compared with the NC1 and PC2 groups, blood glucose of DG1, DG2, and DG3 groups at 60 min and 120 min showed significant increase (*p* < 0.05). The significant results of PC2 at 60 min may be due to glucose in a normal diet. Compared with the DG1 group, both the groups (DG2 and DG3) showed good glycemic control with highly significant differences at 60 min and 120 min as shown in Table 5. Results clearly showed that both the groups on medication (DG2 and 3) had better controlled blood sugar than DG1 at all times, proving the efficacy of both commercial and nanomedicine. Overall, all the groups have significantly different readings than the control, showing the diversity of each group selected for the study, hence the comparison of nano and commercial medicine can be accurately distinguished in the present trial.

### 3.6. TC, TG, LDL-C, and HDL-C

As shown in Table 6, serum TC, TG, and LDL-C in all the diabetic groups (DG1–3) were significantly increased compared with both the control groups (*p* < 0.01). Compared with the DG1 group, all the parameters—except HDL-C in DG2 and DG3—were significantly lower (*p* < 0.05, Table 7). HDL-C is good cholesterol and should be higher in normal cases.

## 4. Discussion

Medicinal plants have been used to treat various human diseases for centuries. Herbs are a potential source of antidiabetic medications, as evidenced by traditional knowledge of the plants use. Secondary metabolites found in medicinal plants—such as flavonoids, terpenoids, alkaloids, and polysaccharides—have been widely researched for their antidiabetic effect. Alkaloid compounds isolated from *C. roseus* have been known to have antidiabetic activity [33,34,35,36,37].

The crude extract of *C. roseus* was used in this study to synthesize silver nanoparticles and to study their cytotoxicity effect on Vero cell lines. The *C. roseus* extraction was completed in a methanolic solvent, also used for phytochemical extractions. The use of methanol has been shown to have higher extraction efficiencies for the phytochemical compounds from herbal plants in the previous study [38,39]. Phenolics and alkaloid compounds were extracted from dry *C. roseus* plant samples in a methanolic solvent in a previous study [40]. The quantitative phytochemical assessment of the various extracts of *C. roseus* revealed methanolic extract as the best solvent to extract the phytochemical compounds. These findings agree with previous studies [41,42] that reported high alkaloid compounds and antidiabetic activity.

*C. roseus* has a lot of medicinal effects; hence, many studies have previously been conducted on the biosynthesis and characterization of silver nanoparticles using *C. roseus* leaf extract and its effects on the cancer cell lines. Similarly, alkaloid compounds present in herbal plants have been reported to have anti-diabetic properties in the previous findings [43]. However, the use of plants as nano-extract has not been previously carried out in these studies in relation to diabetes. Furthermore, the acquired results also indicated that higher cell mortality occurs at higher concentrations of nanoparticles; these results are also supported by previous studies in other diseases apart from diabetes [44].

For curing lethal diseases with huge global impact, long-term effective technique with fewer (or less intense) side-effects will be required with a combined methodology instead of a short-range approach and a targeted way out instead of conventional agriculture approaches for example nanotechnology [2,45]. Therefore, in the present study, plant extract in the green synthesis of nanoparticles was investigated with cell lines and also their use against diabetes [46]. For this purpose, the *C. roseus* extract was added to the 1 mM AgNO_3_ solution, which reduces silver ions into silver particles, as indicated by the change in color from light green to dark brown in the synthesis, which confirmed synthesis of *C.* AgNPs in this study. The solution’s dark color was exhibited due to the surface plasmon resonance phenomenon [47]. Earlier studies indicated that the reduction of silver ions and stability of AgNPs are due to the presence of phyto-constituents or plant metabolites [48,49]. In the current study, the synthesized silver nanoparticles were spherical in shape and 5 μm in size as indicated in SEM and 65 nm AFM images. According to various studies, the size of the spherical-shaped silver nanoparticles synthesized was 35–55 nm, which is known to be an optimum range for cell permeability [50]. The XRD technique is also used to determine the material’s crystal structure and chemical composition, with the diffraction peaks obtained indicating that silver nanoparticles are synthesized [51,52]. The XRD graph was plotted by intensity vs. 2θ. In the spectrum, the sharp peaks show crystalline [13,14] in which atoms were arranged in a periodic pattern of three dimensions (3D). The variation of peaks in XRD spectrum was due to doping or foreign atoms capping, so it confirmed the synthesis of the composite [3]. Furthermore, for the confirmation of the formation of composite capping with Ag metal, EDX analysis was performed. During the EDX measurement, different areas were focused upon and corresponding peaks were observed. Ag metal was seen in the synthesized composite nanostructure in the EDX spectrum [53,54,55].

Synthesis of AgNPs occurred due to interaction of the negative group (*C*OO^−^) or polar group (OH) and tended to attach with Ag+, therefore contributing to reduction and stabilization of Ag ions [52]. FT-IR spectroscopy was used to examine the chemical composition of the surface and the local molecular environment, reducing and capping agents of the silver nanoparticles [56]. In the current study, the reduction of the silver nanoparticles of *C. roseus* was also confirmed by the FT-IR analysis and zeta potential. These findings were under previous studies that characterized silver nanoparticle through FT-IR and zeta potential. *C. roseus* silver nanoparticles were further tested to evaluate their cytotoxicity effect in Vero cells. Dose-dependent cell toxicity was observed at a high concentration of *C.* AgNPs. The low concentration of *C.* AgNps recommended for diabetes drug delivery and the high concentration of nanoparticles can be correlated with anticancer activity of the *C.* AgNPs, because results clearly indicated that a higher level of NP—i.e., 1000 μg/mL was able to kill 69% of the cell (labeled as cytotoxicity level) with 5% cell proliferation. The amount is less in terms of cell proliferation, but the main goal in cancer is to kill or stop damaged/cancerous cells which are verified in the present study. Hence, these NPs work against diabetes but can also play a vital role in the eradication of cancer. These results also support the previous findings by taking into account the NPs at A3 concentration which have less cell toxicity and more cell proliferation, meaning this is the best-suited concentration for the cell lines as cell growth of normal cells was more—i.e., 70%—while cell toxicity was quite low—i.e., 37% [57]. The study results are recommended for the *C.* AgNPs treatment against diabetes. We observed the living and dead cells by using different concentrations of *C.* AgNPs as exposure to high concentrations of silver nanoparticles may generate the reactive oxygen species (ROS) that may result in cytotoxicity of host cells [58,59,60,61,62,63].

In a previous study, *Opuntia* spp. *cacti* also demonstrated anti-hyperglycemic effects or anti-diabetic properties [53,54,55,63]. However, the results were not significant in terms of ‘anti-hyperglycemic’ effects, though the definitive evidence was conflicting [64] and it is important to clarify these health benefits due to the increasing need for prevention and treatment of chronic diseases; therefore, the present study was designed to use these plants in the shape of nanoparticles for its effectiveness and to know the exact role in diabetes. It was also well known that blood glucose and lipid metabolism is inseparable [53,62,63]. However, the nanomedicine developed in this study is quite effective in lowering or maintaining the sugar content in the body.

The results of nanoparticles made from *C. roseus* show a significant effect on water and food intake, rat weight, and urine volume along with blood sugar. This clearly indicates that the nano-extract from this plant is nearly at par with commercially available medicine. Similar findings were also observed in previous studies [59], though their results were slightly different for rat weight with non-significant variations. Similarly, a 200 mg/kg/day of rhamnogalacturonan showed decreased glucose tolerance in streptozotocin-induced diabetic rats [60]. The serum triglyceride levels in high-fat-diet-induced obese rats were significantly ameliorated by ethanol and the synthesis of silver nanoparticle extract of *C. roseus* [61].

## 5. Conclusions

In the present study, silver nanoparticles (AgNPs) from *C. roseus* plant (*C.* AgNPs) were synthesized, characterized, and investigated for their cell viability and antidiabetic analysis. *C.* AgNPs showed excellent cell viability on vero cell lines due to low cytotoxic effects. The antidiabetic potential of *C.* AgNPs on a mammalian model, such as rats, also showed adequate efficacy and improved important indicators in diabetic rats. Low cytotoxic effects and enhanced antidiabetic features could make it a potential candidate for the efficient treatment of diabetes.

## Figures and Tables

**Figure 1 molecules-27-06191-f001:**
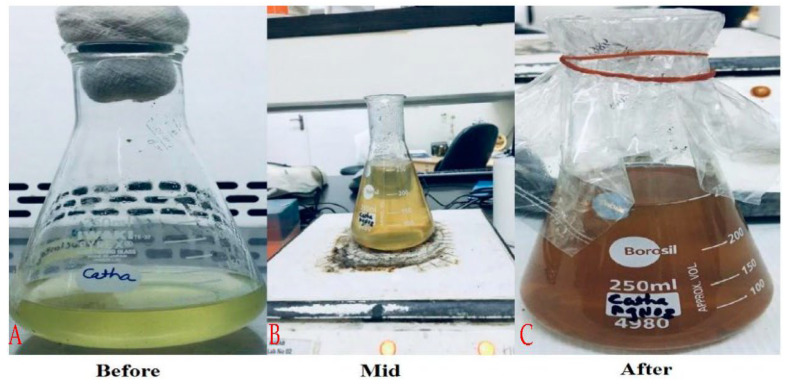
Synthesis of *C. roseus* silver nanoparticles (AgNPs). (**A**) *C. roseus* extract. (**B**) Synthesizing silver-nanoparticles. (**C**) Synthesized silver nanoparticles.

**Figure 2 molecules-27-06191-f002:**
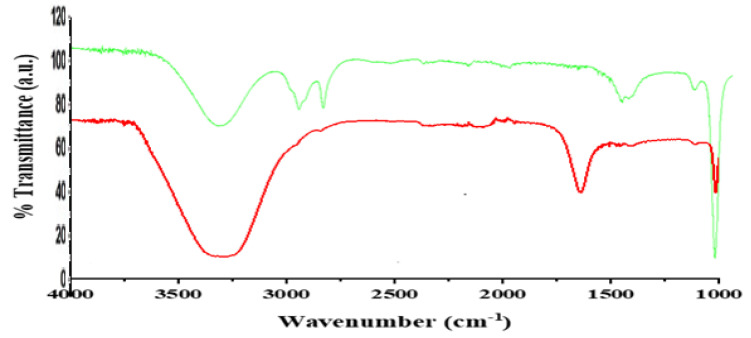
FT–IR spectrum of *C. roseus* leaves extract and silver nanoparticles.

**Figure 3 molecules-27-06191-f003:**
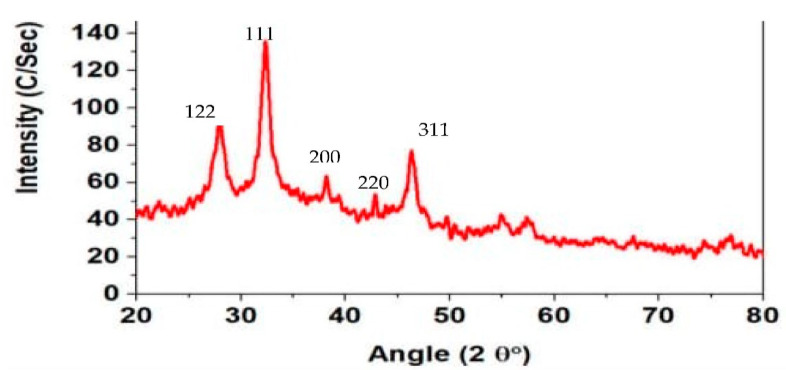
XRD spectrum of silver nanoparticle of the *C. roseus* plant. The XRD peaks of *C. roseus* AgNps at 2θ values of 28.00°, 32.00°, 38.00°, 44.00° and 45.00° represent the planes of silver at 122, 111, 200, and 220 respectively. These planes are matched by standard powder diffraction card of JCPDS, silver file no. 04-0783. The XRD results confirmed the formation of crystalline *C. roseus* AgNps which was matched by silver planes as shown in the figure.

**Figure 4 molecules-27-06191-f004:**
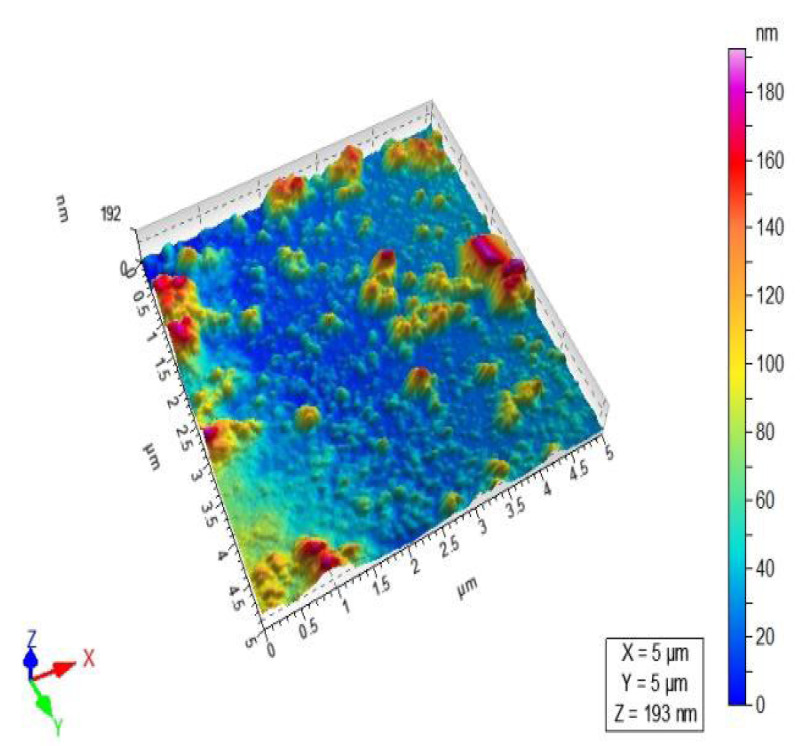
AFM 3D image of *C. roseus* nano particles by using a scale of 60 nm.

**Figure 5 molecules-27-06191-f005:**
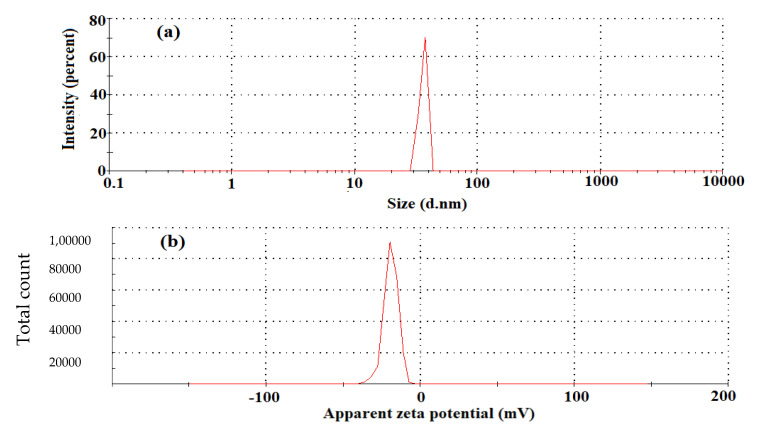
Graph showing (**a**) the DLS at 35 nm ± 10 particle size; and (**b**) zeta potential value determined at −19.2 mV.

**Figure 6 molecules-27-06191-f006:**
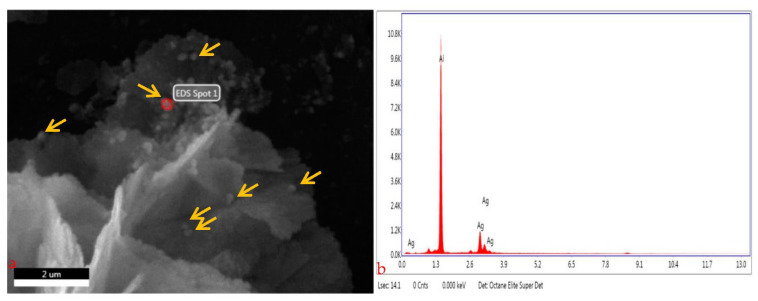
SEM–EDX measurement of different areas showing corresponding peaks (the red spot and the yellow arrows in (**a**) shows the peak on the surface while (**b**) shows the graphical representation of the nanoparticles).

**Figure 7 molecules-27-06191-f007:**
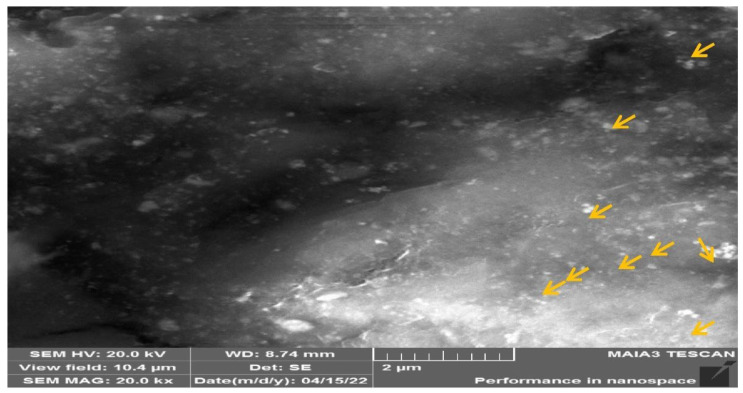
TESCAN shows the even shape and spherical nature of the developed particle (yellow arrows indicate spherical shaped nanoparticle).

**Figure 8 molecules-27-06191-f008:**
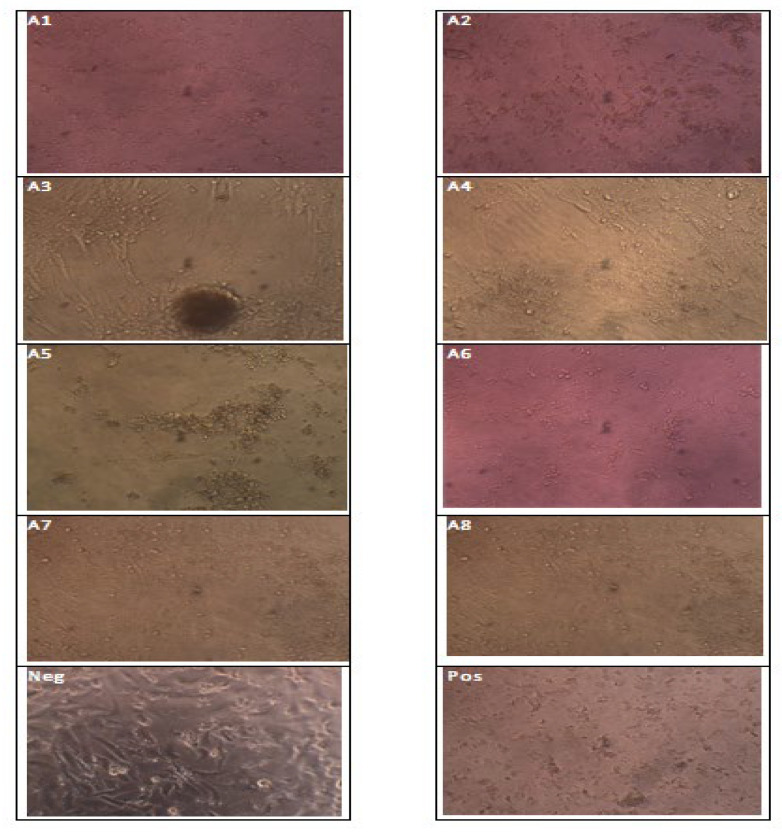
Effects of nanoparticles on the morphology of Vero cells after 14 days. A1 = 142, A2 = 284, A3 = 426, A4 = 568, A5 = 710, A6 = 852, A7 = 970, and A8 = 1000 are serially diluted nanoparticles. The negative control showed no change in morphology of the monolayer, whereas positive control cells were showing classical cytopathic effects.

**Figure 9 molecules-27-06191-f009:**
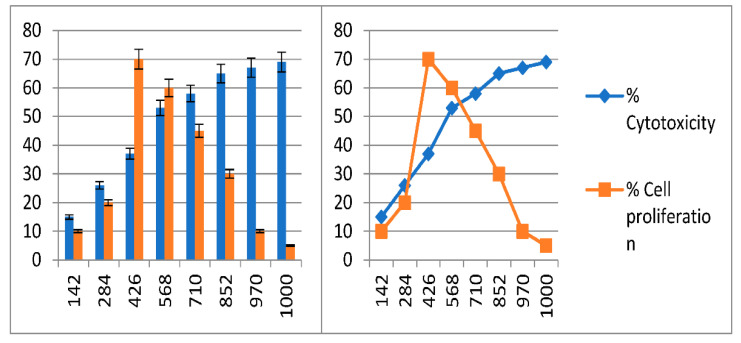
Comparison of cytotoxicity and cell proliferation (*n* = 5) level against various nanoparticle concentrations on cell lines.

**Figure 10 molecules-27-06191-f010:**
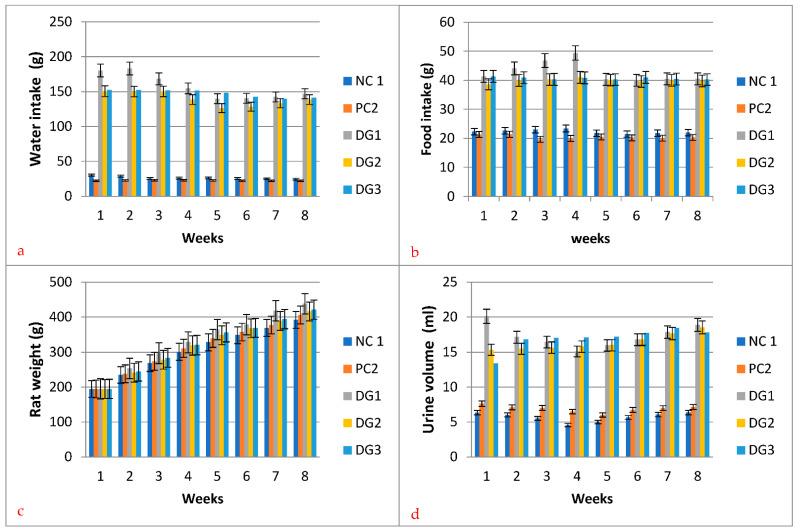
Effect of the nano-aqueous extract *C. roseus* and commercially available medicine (with *n* = 10 for each group) on (**a**) water intake, (**b**) food intake, (**c**) rat weight, and (**d**) urine volume.

**Table 1 molecules-27-06191-t001:** Functional groups of *C. roseus* extract identified by FT–IR.

Origin	Group Frequency	Peak Value	Class
C=C bending	895–885 cm^−1^	890	Alkene
C=C bending	1400–1000 cm^−1^	1000	Alkene
C-OH	1420–1330 cm^−1^	1360	Alcohol
C-H stretching	3000–2840 cm^−1^	2850	Alkane
C-H stretching	3000–2840 cm^−1^	3000	Alkane
C=O	3333–3267 cm^−1^	3333	Carbonyl group
N-H stretching	3500–3300 cm^−1^	3500	Amide

**Table 2 molecules-27-06191-t002:** AFM images of *C. roseus* nanoparticle by 60 scale.

Serial No.	Compound	Scale (nm)	3D Axis	µm	Nanoparticle Size
1	Vindoline	0–60	X	2.64	45 nm
			Y	2.64	
			Z	66.4	

**Table 3 molecules-27-06191-t003:** Values of weight and atomic percent by SEM EDX.

S. No	Element	Weight %	Atomic %
1	AlK	73.34	91.66
2	AgL	26.66	8.34

**Table 4 molecules-27-06191-t004:** Analysis of variance (ANOVA) for eight concentrations of NP with cytotoxicity and cell proliferation.

Source of Variation	SS	df	MS	F	*p*-Value	F Crit
Treatments	9359.993	7	4679.996	15.80039	0.000109 **	3.554557
Concentration	3361.97	1	3361.97	11.1601	0.012408	5.591448
Error	2108.744	7	301.2491			
Total	7606.583	15				

Note: *p*-value with ** indicate highly significant variations between the studied concentration.

**Table 5 molecules-27-06191-t005:** Urine analysis showing glycosuria in the studied groups using urine dipstick test.

Week	NC 1	PC 2	DG 1	DG 2	DG 3
1	−	−	++	+	−
2	−	−	++	−	+
3	−	−	++	−	−
4	−	−	+++	+	−
5	−	−	+++	+	+
6	−	−	+++	+	−
7	−	−	+++	−	−
8	−	−	+++	−	−

Note: negative sign represents no sugar level, + is mild sugar level, ++ moderate sugar levels, and +++ represents higher level of sugar in the urine.

**Table 6 molecules-27-06191-t006:** Glucose tolerance effect in glucose-induced hyperglycemia.

Group	30 min (mmol/L)	60 min (mmol/L)	120 min (mmol/L)
NC1	4.2	3.9	3.7
PC2	6.1	8.2 *	7.3
DG1	12.7 *	13.8 **	13.1 **
DG2	7.1	9.3 *	8.2 *
DG3	7.4	9.7 *	8.8 *

Note: Data were compared with NC1 group. Highly significant differences are represented by “**” while “*” is used for significant differences.

**Table 7 molecules-27-06191-t007:** Effects of aqueous extract of *C.*
*roseus* on serum TC, TG, LDL-C, and HDL-C in diabetic rats.

Group	TC (mmol/L)	TG (mmol/L)	LDL-C (mmol/L)	HDL-C (mmol/L)
NC1	1.71	0.89	1.01	0.41
PC2	1.79 *	0.94 *	1.11 *	0.46
DG1	2.72 **	2.21 **	1.21 *	0.31
DG2	2.52 *	1.36 *	0.71	0.39
DG3	2.61 *	1.49 *	0.86	0.38

Note: Data were compared with NC1 group. Highly significant differences are represented by “**” while “*” is used for significant differences.

## Data Availability

The data presented in this study are available on request from the corresponding author.

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
