# Peer review of "Biogenic Synthesis of Silver Nanoparticles Using *Catharanthus roseus* and Its Cytotoxicity Effect on Vero Cell Lines"

_molecules, 2022, doi:10.3390/molecules27196191_

Round 1
Reviewer 1 Report (Previous Reviewer 3)
The authors have followed all suggestions and comments. The manuscript can now be published.
Author Response
Reviewer 1
We are highly thankful and appreciable to the reviewer’s 1 for his encourageous and positive remarks on our manuscript. We look forward for the acceptance and publication of our manuscript as per reviewer’s 1 comments.
Reviewer 2 Report (Previous Reviewer 2)
1. Fig. 2 and 3 should be merged as one figure so that FTIR spectra of the plant extract and the nanoparticles could be done.
2. “Group II positive control (PC2): normal rats treated with 510 mg/kg b.w./d of silver nanoparticle with aqueous Catharanthus extract(n=10)” The nanoparticles treated mouse can’t be considered as a positive control. This is a wrong positive control.
3. “ The rats used in the pre- Molecules 2022, 27, x FOR PEER REVIEW 5 of 19 sent study were approved by the Animal Ethics Committee of the National Institute of genomics and advanced biotechnology (Pakistan).” Authors should provide approval no. for conducting studies.
4. The FTIR data need to be analyzed again The authors should describe the shift in the vibrational peaks of plant extract when silver nanoparticles are formed.
5. The XRD pattern of the silver NPs should be in accordance with the JCPDS file of the nanosilver. This need to be mentioned in the manuscript
6. The abbreviations should be described when used the first time in the manuscript after that there is no need to write the complete word, only abbreviations can be given.
7. In fig. 5 the scale is not visible, different panels of the figure should be labeled as A, B, C, etc.
8. What is the relevance of the negative charge in context to physiology? please discuss.
9. The sem image Fig. 7, is not up to the mark, The nanoparticles are looking like a sheet, why is this morphology?
10. “The quantity of AlK and AgL was 73.34, and 26.6 respectively” What is Alk, and what are the units of this quantity?
11. Fig 8 is not publication quality and what is the size of the nanoparticles according to electron microscopy?
12. Fig 9, not up to the mark. How IC 50 was calculated and the analysis of the cytotoxicity assay is very confusing. Why this range of concentration was chosen?
13. The English language is inadequate, and the animal experiments need to be explained properly.
Author Response
Reviewer 2:
All the mentioned paper and comments have been successfully incorporated and addressed in the revised manuscript. Followings are the one by one response of each comment for the revision of this manuscript.
- 2 and 3 should be merged as one figure so that FTIR spectra of the plant extract and the nanoparticles could be done.
Comment: We appreciate the constructive suggestion of reviewer and we have merged both Figures 2 and 3 in one figure i.e. Figure 2 in the revised manuscript.
- “Group II positive control (PC2): normal rats treated with 510 mg/kg b.w./d of silver nanoparticle with aqueous Catharanthus extract(n=10)” The nanoparticles treated mouse can’t be considered as a positive control. This is a wrong positive control.
Comment: we appreciate the reviewer for his correction. The nanoparticle word was added by mistake. It has been removed. PC2 was only given Catharanthus extract (as a normal food) without loaded with silver nanoparticle. The correction has been made and incorporated in the revised manuscript.
- “ The rats used in the pre- Molecules 2022, 27, x FOR PEER REVIEW 5 of 19 sent study were approved by the Animal Ethics Committee of the National Institute of genomics and advanced biotechnology (Pakistan).” Authors should provide approval no. for conducting studies.
Comment: Thank you very much for the reviewer’s comment. The approval number for the conducting study was 528/Reg./MICB/2022.
- The FTIR data need to be analyzed again. The authors should describe the shift in the vibrational peaks of plant extract when silver nanoparticles are formed.
Comment: We appreciate this positive comment and we have added the description of FTIR in the revised manuscript.
- The XRD pattern of the silver NPs should be in accordance with the JCPDS file of the nanosilver. This need to be mentioned in the manuscript
Comment: we have incorporated these changes in the revised manuscript.
- The abbreviations should be described when used the first time in the manuscript after that there is no need to write the complete word, only abbreviations can be given.
Comment: The comment has been addressed and all such changes have been made in the revised manuscript.
- In fig. 5 the scale is not visible, different panels of the figure should be labeled as A, B, C, etc.
Comment: We have updated all such changes in the revised form of article. The scale is visible now and all panels have been labeled as A and B in the updated article.
- What is the relevance of the negative charge in context to physiology? please discuss.
Comment: The negative charge relevance has been discussed in the manuscript.
- The sem image Fig. 7, is not up to the mark, The nanoparticles are looking like a sheet, why is this morphology?
Comment: The graphical representation (arrows indication nanoparticle) has been incorporated in the revised form.
- “The quantity of AlK and AgL was 73.34, and 26.6 respectively” What is Alk, and what are the units of this quantity?
Comment: AlK is the Aluminium peak in EDX. Mostly we take the percentage of this quantity while electron beams on the EDX are taken which is represented in the units of in KeV. We have added this information in the revised manuscript.
- Fig 8 is not publication quality and what is the size of the nanoparticles according to electron microscopy?
Comment: The figure has been replaced with the new one in the revised manuscript.
- Fig 9, not up to the mark. How IC 50 was calculated and the analysis of the cytotoxicity assay is very confusing. Why this range of concentration was chosen?
Comment: We have removed this confusion and all the comparisons of in vitro trials have been explained graphically in the mentioned figure. Furthermore, the calculation of IC50 has been explained.
- The English language is inadequate, and the animal experiments need to be explained properly.
Comment: we have revised the scientific language and explained the animal model experimentation of our manuscript as per reviewer’s comment.

Reviewer 3 Report (Previous Reviewer 1)
Khansa Jamil and co-workers submitted the revised manuscript entitled “Biogenic synthesis of silver nanoparticles using Catharanthus roseus and its cytotoxicity effect on vero cell lines Earlier version: molecules-1763299” to be published in “Molecules (I. F = 4.927)”. In this paper, they describe the importance of biogenic synthesized silver nanoparticles towards diabetes research. Author improved the manuscript as per recommendation of reviewers, it can be accepted after addressing the queries.
1. In the introduction cite following relevant literature on biogenic Ag NPs synthesize with certain unique applications; 1. Surfaces 2022, 5, 67-90; 2. Chem. Eng. J. 2021, 407, 127202; 3. Nanomaterials 2022, 12, 830.
2. Assign the XRD peaks with respect to available JCPDS file.
3. In section 3.2.4. Zetasizer & Zetapotential, write the sentence as “The zeta potential value was determined as -19.2 mV, a measure of the stability of the nanoparticles” and mention the DLS value as 35 ± ? nm (the standard deviation must be mentioned in place of ?).
4. For Cytotoxicity plots in Fig.10, author must provide error bars from at least 3 sets of measurements and n=3 must be mentioned in the caption as well.
5. Figure number 8 is wrong, it should be numbered as Figure 11. Also, in the caption number of data (n =?) considered for the error bars should be mentioned. Follow the uniformity for writing the Figures or Fig.
6. Merits of the research still not sufficiently justified in the conclusion part.
Author Response
Reviewer 3:
- In the introduction cite following relevant literature on biogenic Ag NPs synthesize with certain unique applications; 1). Surfaces 2022, 5, 67-90; 2). Chem. Eng. J. 2021, 407, 127202; 3). Nanomaterials 2022, 12, 830.
Comment: All the mentioned articles have been incorporated in the revised manuscript and highlighted as well.
- Assign the XRD peaks with respect to available JCPDS file.
Comment: All the peaks have been assigned in the XRD analysis.
- In section 3.2.4. Zetasizer & Zetapotential, write the sentence as “The zeta potential value was determined as -19.2 mV, a measure of the stability of the nanoparticles” and mention the DLS value as 35 ± ? nm (the standard deviation must be mentioned in place of ?).
Comment: we have updated this information in the revised manuscript.
- For Cytotoxicity plots in Fig.10, author must provide error bars from at least 3 sets of measurements and n=3 must be mentioned in the caption as well.
Comment: Basically 5 sets of trials were done for cytotoxicity analyses and it has been mentioned and highlighted in the legend of figure 9.
- Figure number 8 is wrong, it should be numbered as Figure 11. Also, in the caption number of data (n =?) considered for the error bars should be mentioned. Follow the uniformity for writing the Figures or Fig.
Comment: We have updated and corrected these ambiguities in the revised manuscript.
- Merits of the research still not sufficiently justified in the conclusion part.
Comment: we have updated the conclusion part and added key information in this section of revised manuscript.

Reviewer 4 Report (New Reviewer)
Authors synthesized silver nanoparticles using Catharanthus roseus and checked its cytotoxicity effect on Vero cell lines.
1. Previously various studies reported the synthesis of silver nanoparticles including Catharanthus roseus so what is the novelty of this work?
2. Authors should include the importance of green synthesis in the introduction.
3. Authors should include some more references in the introduction some of them are
Roy, A., Sharma, A., Yadav, S., Jule, L. T., & Krishnaraj, R. (2021). Nanomaterials for remediation of environmental pollutants. Bioinorganic Chemistry and Applications, 2021.
Mittal, S., & Roy, A. (2021). Fungus and plant-mediated synthesis of metallic nanoparticles and their application in degradation of dyes. In Photocatalytic degradation of dyes (pp. 287-308). Elsevier.
Roy, A., Singh, V., Sharma, S., Ali, D., Azad, A. K., Kumar, G., & Emran, T. B. (2022). Antibacterial and Dye Degradation Activity of Green Synthesized Iron Nanoparticles. Journal of Nanomaterials, 2022.
4. Scientific names should be in italics
5. Why figures 2 and 3 are in different forms they should be in the same format or rather make in one figure for better comparison
6.
Author Response
Reviewer 4:
- Previously various studies reported the synthesis of silver nanoparticles including Catharanthus roseus so what is the novelty of this work?
Comment: The novelty of this study is the key application of Catharanthus roseus nanoparticles on Vero cell lines. The in vitro cytotoxic analyses on vero cell line and mammalian model have not been reported in the previous studies. Moreover, in the present study C. roseus nanoparticles (AgNp) were also compared with the commercially used antidiabetic drugs.
- Authors should include the importance of green synthesis in the introduction.
Comment: we have incorporated the importance of green synthesis of nanoparticles in the updated introduction.
- Authors should include some more references in the introduction some of them are
Roy, A., Sharma, A., Yadav, S., Jule, L. T., & Krishnaraj, R. (2021). Nanomaterials for remediation of environmental pollutants. Bioinorganic Chemistry and Applications, 2021.
Mittal, S., & Roy, A. (2021). Fungus and plant-mediated synthesis of metallic nanoparticles and their application in degradation of dyes. In Photocatalytic degradation of dyes (pp. 287-308). Elsevier.
Roy, A., Singh, V., Sharma, S., Ali, D., Azad, A. K., Kumar, G., & Emran, T. B. (2022). Antibacterial and Dye Degradation Activity of Green Synthesized Iron Nanoparticles. Journal of Nanomaterials, 2022.
Comment: All the references have been added in the revised manuscript
- Scientific names should be in italics
Comment: All the corrections have been made as per reviewer’s suggestion.
- Why figures 2 and 3 are in different forms they should be in the same format or rather make in one figure for better comparison
Comment: Both figures have been merged as Figure 2 for comparison in the revised manuscript.

Round 2
Reviewer 2 Report (Previous Reviewer 2)
The manuscript still needs a lot of improvement.
1. The quality of the figures needs to be improved. the font size of the axis, legends, and labeling in all figures should be the same. Authors should read the journal guidelines before submitting the paper.
2. The ethical approval number for the conducting study should be incorporated in the manuscript.
3. The in vivo relevance of the negative charge of the particles is still missing.
4. Figure numbering is wrong and fig 8 of electron microscopy is missing.
5. AlK is the Aluminium peak in EDX and should be incorporated in the manuscript.
6. The English language still needs a lot of improvement.
Author Response
Reviewer 2
- The quality of the figures needs to be improved. The font size of the axis, legends, and labeling in all figures should be the same. Authors should read the journal guidelines before submitting the paper.
Comment: we appreciate for pointing out the figure related issues. All the font size, labeling, etc. has been done in according to the journal guidelines. The figure quality has been also improved, but some of the pictures were taken from software so further quality improvement is not possible, it is therefore requested to consider the picture in this improved version of the manuscript
- The ethical approval number for the conducting study should be incorporated in the manuscript.
Comment: Ethical approval number has been incorporated in the revised manuscript.
- The in vivo relevance of the negative charge of the particles is still missing.
Comment: Thanks for your suggestion, our prime objective is not the mechanism of nanoparticles in in-vivo study, our prime goal in this study is to check the viability cell proliferation. But still we have incorporated the relevance. The presence of electrons (negative surface charge which may be attributed to deprotonated OH groups) at the silver nanoparticle surface which became more significant with reducing the particle size made them negatively charged. Nanoparticles with negative charge are less toxic in in vivo animal model as compared to the positive charged particles.
For further explanations in the comment section: The boundary between the solid and the fluid phase is a dynamic environment, and multiple phenomena, such as the presence of dangling bonds, or the adsorption or grafting of charged molecules contribute to the appearance of a net charge on the nanoparticle surface. This charge has a primary effect on the behavior of nanoparticles in different environments, in particular on controlling their tendency toward aggregation, as electrostatic repulsion between particles which is a key factor promoting the stability of colloidal solutions. In particular, in an electrolyte solution, mobile charges in solution are attracted by the static charges on the nanoparticle surface, effectively leading to a screening of the electric potential, which can ultimately result in particle aggregation. A typical measure of surface charge and colloidal stability is given by the zeta potential, which is defined as the electric-potential difference between the stationary layer of charges surrounding the particles and the solution potential
- Figure numbering is wrong and fig 8 of electron microscopy is missing.
Comment: the figure number has been corrected but there is no missing figure it may be the MS word compatibility issue as we have downlaed the manuscript from mdpi website and the figure was present
- AlK is the Aluminium peak in EDX and should be incorporated in the manuscript.
Comment: The EDX spectrum also revealed the AgNPs synthesis. The composition of AgNPs elements analyzed Al and Ag by the spectrum .Metallic silver nano crystals with optical absorption peaks approximately at 1.3 keV for Al and 2.8 keV for Ag respectively, due to surface plasmon resonance.
- The English language still needs a lot of improvement.
Comment: The English language has been significantly improved in the revised manuscript.

Reviewer 3 Report (Previous Reviewer 1)
Author rectified all the pointed issues, it can be accepted in its current form.
Author Response
Reviewer 3
We greatly appreciate the reviewer’s positive consent for the acceptance of this manuscript. The suggestion raised by reviewer about language is quite constructive and we have improved the language of whole manuscript which will definitely improve this manuscript in its presentation.

Round 3
Reviewer 2 Report (Previous Reviewer 2)
All the comments have been addressed except the quality of the figures.
This manuscript is a resubmission of an earlier submission. The following is a list of the peer review reports and author responses from that submission.
Round 1
Reviewer 1 Report
Khansa Jamil and co-workers submitted the manuscript entitled “Biogenic synthesis of silver nanoparticles using Catharanthus roseus and its cytotoxicity effect on vero cell lines” to “Molecules (I. F = 4.412)”. In this paper, they wish to describe the importance of biogenic synthesized silver nanoparticles. But, due to lacks in presentation, supportive evidences and applications, it looks like a premature manuscript.
1. Introduction written in a poor manner describing the diabetes. In fact, there are no anti-diabetic applications derived from the material reported in this manuscript. Merits of biogenic synthesize of silver nanoparticles must be pronounced. Based on the provided introduction, no one can understand about the main resolved problem in this manuscript.
2. Characterization part is poorly presented and explained. There is no assign of diffraction pattern for XRD peaks to suggest the formation of Ag NPs.
3. Where are the TEM, DLS and EDX data to represent the formation of Ag NPs, particle size and distribution details. AFM is not properly explained, which may give approximate particle size. TEM is an essential data.
4. There is no MTT assay bar plot and IC50 plot, but mentioned in the text. Actually, presentation is not perfect to claim this publication. Nanoparticles effect on vero cell lines is not sufficient to say the anti-diabetic potential.
5. If the author wishes to justify the nanoparticle’s anti-diabetic potential, solid evidences such as mice studies / clinical trials are essential.
6. Conclusion section must contain the merits and limitation of the biogenically synthesized Ag NPs. Currently, it is looks like a normal one exploring more on anti-diabetic potential, which does not have any evidence.
Reviewer 2 Report
This manuscript is presented very badly. There is no relation between the experiments done and Type-2 diabetes mellitus (T2DM). Therefore this work is not suitable for publication.
1. First of all it is not a novel work, the silver nanoparticles of Catharanthus roseus have been synthesized by many labs, and there are like 4-6 publications on this subject.
Please refer to these papers: https://doi.org/10.1016/j.tiv.2020.104910, DOI: 10.4172/2157-7439.1000305, DOI: 10.4172/2157-7439.1000305
2. The authors are discussing type 2 diabetes, but there is no data that shows anti-diabetic activity.
3. The presentation of the paper is not up to the marks, like no paragraphs, and sections are entangled.
4. There are many sentences that do not make any sense.
Reviewer 3 Report
The authors Sahir Hameed Khattak, Khansa Jamil, Muhammad Naeem Riaz, Sania Begum, Anum Farrukh, Aish Muhammad, Touqeer Taj, Imran Khan, Sundus Riaz, Huma Batool, Sabahat Majeed, Sajid Ali Khan Bangash, Alia Mushtaq, Ghulam Muhammad Ali, Shahab Bashir, Imdad Kaleem have submitted a manuscript (ID: molecules-1763299) entitled: “Biogenic synthesis of silver nanoparticles using Catharanthus roseus and its cytotoxicity effect on vero cell lines” to the section: Materials Chemistry to the scientific journal Molecules. I have carefully read the draft that it would fit as improved article into the section of this journal.
In order to improve the manuscript the following article has to be discussed in detail in the text-part of the draft:
Krishna G, Srileka V, Singara Charya MA, Abu Serea ES, Shalan AE. Biogenic synthesis and cytotoxic effects of silver nanoparticles mediated by white rot fungi. Heliyon. 2021;7(3):e06470. Published 2021 Mar 16. doi:10.1016/j.heliyon.2021.e06470
Furthermore, it makes sense to correlate the findings discussed in the draft with those published for prickly pears:
Gouws CA, Georgousopoulou EN, Mellor DD, McKune A, Naumovski N. Effects of the Consumption of Prickly Pear Cacti (Opuntia spp.) and its Products on Blood Glucose Levels and Insulin: A Systematic Review. Medicina (Kaunas). 2019;55(5):138. Published 2019 May 15. doi:10.3390/medicina55050138